# Bromine Ion-Intercalated Layered Bi_2_WO_6_ as an Efficient Catalyst for Advanced Oxidation Processes in Tetracycline Pollutant Degradation Reaction

**DOI:** 10.3390/nano13182614

**Published:** 2023-09-21

**Authors:** Rama Krishna Chava, Misook Kang

**Affiliations:** Department of Chemistry, College of Natural Sciences, Yeungnam University, 280 Daehak-ro, Gyeongbuk 38541, Gyeongsan, Republic of Korea

**Keywords:** bottom-up synthesis route, Bi_2_WO_6_, morphology tuning, layered structure, advanced oxidation process

## Abstract

The visible-light-driven photocatalytic degradation of pharmaceutical pollutants in aquatic environments is a promising strategy for addressing water pollution problems. This work highlights the use of bromine-ion-doped layered Aurivillius oxide, Bi_2_WO_6_, to synergistically optimize the morphology and increase the formation of active sites on the photocatalyst’s surface. The layered Bi_2_WO_6_ nanoplates were synthesized by a facile hydrothermal reaction in which bromine (Br^−^) ions were introduced by adding cetyltrimethylammonium bromide (CTAB)/tetrabutylammonium bromide (TBAB)/potassium bromide (KBr). The as-synthesized Bi_2_WO_6_ nanoplates displayed higher photocatalytic tetracycline degradation activity (~83.5%) than the Bi_2_WO_6_ microspheres (~48.2%), which were obtained without the addition of Br precursors in the reaction medium. The presence of Br^−^ was verified experimentally, and the newly formed Bi_2_WO_6_ developed as nanoplates where the adsorbed Br^−^ ions restricted the multilayer stacking. Considering the significant morphology change, increased specific surface area, and enhanced photocatalytic performance, using a synthesis approach mediated by Br^−^ ions to design layered photocatalysts is expected to be a promising system for advancing water remediation.

## 1. Introduction

Over the years, pollutants in the environment, such as organic dyes, phenols, and pharmaceutical waste products, have posed severe threats to human health and ecosystems [1,2]. Various treatment processes, such as membrane filtration [3,4], biological degradation [5,6], and adsorption methods [7,8], have been used to remove pollutants from aquatic environments. Advanced oxidation processes (AOPs) have been proposed as a promising method that generates active radicals (•OH, •O^2−^) that can effectively degrade organic pollutants through redox reactions [9,10]. However, these AOPs usually require electrical energy and external oxidants, which worsen the energy crisis to a certain extent [11,12]. Recently, the semiconductor-based photocatalytic degradation of organic/pharmaceutical pollutants using natural sunlight has attracted significant attention, owing to the formation of abundant reactive radicals during the degradation process [13,14]. Therefore, researchers have made several efforts to synthesize visible-light-active stable photocatalysts, such as CdS [15], g-C_3_N_4_ [16], Bi_2_MoO_6_ [17], and Bi_2_WO_6_ [18].

Among the various photocatalysts, Bi_2_WO_6_ is generally considered a promising and stable material owing to its excellent physicochemical properties, narrow bandgap (~2.8 eV), and band transition from the hybrid orbitals of Bi-6s and O-2p to the W-5d orbitals [19]. Moreover, as a typical Aurivillius oxide, Bi_2_WO_6_ is composed of alternating layers of [Bi_2_O_2_]^2+^ and [WO_4_]^2−^ with the O-atoms sharing, resulting in a perovskite-type oxide [20]. This type of layered structure can expedite the development of an internal electric field and accelerate charge carrier transfer [21]. Notably, it has stimulated photocatalytic applications because of its nontoxicity, low cost, and excellent thermal and chemical stabilities. For example, a simple and reproducible approach to fabricating porous Bi_2_WO_6_ films was designed for the decomposition of methylene blue (MB) under visible light (λ > 420 nm) irradiation, and the resulting porous films showed higher photocatalytic activity than non-porous films [22]. A recent report suggested that doping and heterojunctions in Bi_2_WO_6_ effectively improve the degradation of organic pollutants. The authors compared the photocatalytic activities of Mo-doped Bi_2_WO_6_ and Bi_2_WO_6_/g-C_3_N_4_ composites and confirmed the superiority of the doping strategy [23]. However, the photocatalytic activity of bulk Bi_2_WO_6_ is poor, owing to the rapid electron-hole recombination rate and low number of reactive sites, which results from its low specific surface area. Moreover, the photocatalytic activity of Bi_2_WO_6_ can be enhanced by obtaining different morphologies from various synthesis routes or by imparting a structural hierarchy and porosity [24]. 

Novel three-dimensionally ordered macroporous (3DOM) Bi_2_WO_6_ was successfully prepared by the silica colloidal template method, using Bi(NO_3_)_3_ and phosphotungstic acid as precursors. The derived 3DOM Bi_2_WO_6_ exhibits excellent photocatalytic degradation of phenol and ammonia under solar irradiation [25]. Subsequently, a 3D hierarchical Bi_2_WO_6_ nanostructured photocatalyst was developed using a hydrothermal method, in which the pH significantly affected the morphology of Bi_2_WO_6_. In this work, the authors prepared Bi_2_WO_6_ at pH values ranging from 6 to 12 and achieved morphological changes from nanosheets to hierarchical spheres, which showed high activity for the photocatalytic degradation of MB [26]. A facile microwave hydrothermal route was also utilized to synthesize monodisperse 3D hierarchical Bi_2_WO_6_ microspheres, which were used as catalysts for the photooxidation of NO gas in air under visible-light irradiation [27]. These Bi_2_WO_6_ microspheres show excellent photocatalytic activity and stability. The size and morphology of the prepared Bi_2_WO_6_ nanostructures play a major role in enriching their photocatalytic activity. Changes in the morphology of Bi_2_WO_6_ can increase the exposure of reactive sites, thus increasing pollutant degradation activity [28,29]. Bi_2_WO_6_ nanocrystals with rich oxygen vacancies have been used to build photoelectrochemical sensors for the efficient, fast, and wide-ranging detection of H_2_O_2_. The H_2_O_2_ detection limit reaches 0.5 μM, based on the oxygen vacancy in the Bi_2_WO_6_-nanocrystal, which is ten times higher than that achieved without light irradiation. In this study, the authors confirmed that a narrower band gap of 1.71 eV contributes to better visible light harvesting behavior and increased oxygen vacancies, and can stimulate the exposure of active sites, leading to a high current response and low reduction peak potential for the catalytic reduction of H_2_O_2_ [30]. Bi_2_WO_6_ nanostructures obtained by different synthesis methods have different morphologies, structural properties, and specific advantages and disadvantages. Hence, to overcome the drawbacks associated with Bi_2_WO_6_ nanostructures, it is essential to understand the evolution of various morphologies [31].

Ultrathin nanosheets with a thickness of a few nanometers have been identified as excellent photocatalytic materials because of their rapid mass transport, high charge transfer rate, and an abundance of active sites for catalytic reactions [32]. Unfortunately, converting the morphology of the bulk structure into ultrathin nanosheets is challenging because the intralayer coordination bonds are easily broken during synthesis. Wet bottom-up chemical synthesis has been demonstrated to be the most favorable approach for preparing ultrathin nanosheets with high yields [33,34]. It should also be noted that this synthesis protocol prevents the grown ultrathin nanosheets from stacking together. Therefore, in this study, we propose a single-step hydrothermal method for the preparation of Bi_2_WO_6_ nanostructures and detail the importance of surfactants that alter the morphology from bulk hierarchical microspheres to ultrathin nanosheets and initiate further photophysical changes. These morphological changes, introduced via the organic surfactant-mediated synthesis route, were found to significantly improve the photocatalytic activity by lowering the bandgap, increasing the specific surface area, and improving the charge separation properties. 

In this study, we used a bromine-containing precursor (cetyltrimethylammonium bromide (CTAB)/tetrabutylammonium bromide (TBAB)/potassium bromide (KBr)) assisted bottom-up synthesis to obtain ultrathin nanosheets. The successfully synthesized ultrathin Aurivillius oxide, Bi_2_WO_6_, has a layered structure. Moreover, the Br^−^ ions from CTAB/TBAB/KBr strongly adsorb on the layered surface and block the stacking of layers, as a result, ultrathin Bi_2_WO_6_ nanosheets were obtained, whereas the Bi_2_WO_6_ microspheres were observed without the addition of Br-containing precursors. Consequently, the Br^−^ ions on the surface reduced the bandgap in ultrathin nanosheets. These Bi_2_WO_6_ nanosheets possess a lower bandgap and larger specific surface area than the Bi_2_WO_6_ microspheres. Furthermore, we speculated that the introduced Br^−^ ions might reduce the energy barrier in oxygen vacancy (OV) generation, thus resulting in the gradient concentration of OVs on different exposed facets of materials [30,35]. The photocatalytic activity of the obtained Bi_2_WO_6_ nanostructures was determined by the photocatalytic degradation of the pharmaceutical pollutant tetracycline hydrochloride under visible-light irradiation.

## 2. Materials and Methods

### 2.1. Chemicals and Reagents

Bismuth nitrate pentahydrate (Bi(NO_3_)_3_·5H_2_O; Junsei Chemicals, Tokyo, Japan, 98% purity), sodium tungstate dihydrate (Na_2_WO_4·_2H_2_O; Junsei chemicals, Tokyo, Japan, 99% purity), cetyltrimethylammonium bromide (CTAB, C_19_H_42_BrN, Daejung chemicals, Siheung-si, South Korea, 99% purity), tetrabutylammonium bromide (TBAB, [CH_3_(CH_2_)_3_)]_4_BrN; Daejung chemicals, Siheung-si, South Korea, 98% purity), and potassium bromide (KBr; Junsei Chemicals, Tokyo, Japan, 99% purity) were used without further purification. Distilled water (DI H_2_O) was used for all solutions.

### 2.2. Synthesis of Various Bi_2_WO_6_ Nanostructures (BW NSs)

For the synthesis of bismuth tungstate nanostructures, sodium tungstate and bismuth nitrate were used as the W and Bi precursors, respectively. The detailed procedure for the synthesis of the BW NSs is as follows. First, 0.16 g of sodium tungstate was dissolved in distilled water (40 mL). After stirring for ten minutes, 0.48 g of bismuth nitrate was added, and the mixture was stirred for 30 min. The resultant white precipitated solution was transferred to a 50 mL Teflon-lined stainless steel autoclave and treated at 120 °C for 24 h. Then, the autoclave was cooled, and the obtained product was collected by washing it with DI water three times and drying it in an electric oven at 70 °C overnight. The resultant powder was ground in a mortar for a few minutes, stored in a glass vial, and labeled as BW. Next, the bromine-ion-based surfactant-assisted synthesis procedure for BW NSs was conducted in the same way as the above protocol, the only difference being the addition of 0.025 g of either CTAB or TBAB. The products obtained using CTAB and TBAB were named BW-C and BW-T, respectively. Furthermore, BW-K was prepared separately by adding 9 mg of KBr to the reaction process. 

### 2.3. Physical Characterization Techniques

The morphologies of the samples were analyzed by transmission electron microscopy (TEM; H-7600, Hitachi, Tokyo, Japan) at different magnifications. Field emission-scanning electron microscope (FE-SEM) images were recorded on a Hitachi S-4800 (Hitachi, Tokyo, Japan), The crystal phase and structural information of the prepared BW NSs were analyzed by recording the powder X-ray diffraction patterns (Panalytical XPert Pro diffractometer, 10–80° at Cu-Kα radiation of 1.54060 Ǻ, Malvern Panalytical B.V., Almelo, The Netherlands). Raman spectra were recorded using the Horiba XPlora Plus spectrophotometer (HORIBA Ltd., Kyoto, Japan) in the wavenumber range of 50–1500 cm^−1^. The elemental oxidation states and atomic percentages of the constituent elements in the BW NSs were studied using X-ray photoelectron microscopy (XPS; Thermo-Scientific K-alpha instrument, Thermo Fisher Scientific Inc., Waltham, MA, USA). The valence band edge positions of the resultant samples were also obtained using the same instrument by recording the valence band XPS spectra. Nitrogen adsorption and desorption isotherms of all samples were measured on a BELSorp-II Mini instrument (Microtrac MRB, Haan, Germany. The optical absorption and charge recombination properties of the Bi_2_WO_6_ NSs were studied using a spectrophotometer (Scinco Co. Ltd., Seoul, South Korea) and spectrofluorometer (Scinco Co. Ltd., Seoul, South Korea), respectively. 

### 2.4. Photocatalytic Activity Experiments

The photocatalytic activities of all the BW NSs were determined from the degradation of the pharmaceutical pollutant, tetracycline hydrochloride (TC), in aqueous solutions using a 150 W Xenon lamp with a cut-off filter of 420 nm. Afterward, 15 mg of the prepared BW-K photocatalyst was suspended in TC (100 mL, 20 mg/L) via sonication and stirring for a few minutes. Before light irradiation, the mixed catalyst solution was vigorously stirred in the dark for 30 min to reach adsorption-desorption equilibrium. Next, under visible light irradiation, 5 mL of the solution was removed, filtered, and analyzed at regular intervals. The degradation activity of the photocatalysts was studied by monitoring the absorbance peak of TC at 358 nm, using a Scinco spectrophotometer. The durability of the optimized sample was tested in repeated experiments under the above reaction conditions. In this cycling experiment, the BW-K sample was collected via washing and centrifugation with water several times and dried at 60 °C overnight to obtain the refreshed sample for use in the next cycle. 

## 3. Results and Discussion

### 3.1. Synthesis and Morphological Studies

Figure 1 shows the synthesis method of BW NSs, which includes the control of morphology from microspheres to nanosheets via Br^−^ ion insertion into the layered structure. Without the participation of Br^−^ ions, Bi and W ions will spontaneously self-assemble into a regular hierarchical microsphere-shaped Bi_2_WO_6_ (BW). With the participation of Br^−^- ions from the precursors of CTAB/TBAB/KBr strongly adsorb on the monolayered surface and consequently become negatively charged. During the self-assembly process, monolayer stacking is obstructed by Coulomb repulsion forces and the hydrophobic chains of the CTA^+^ or TBA^+^ ions. The Br^−^ ions on the surface induce a reduction in the bandgap energy of the monolayers [36]. When Br^−^ ions were used in the bottom-up preparation protocol for Bi_2_WO_6_, they would bond to the surface of monolayers to partially dismiss the dangling bonds and create a negatively charged surface, which would reduce the stacking of the layers [36].

The morphological features of the obtained BW NSs were examined using FE-SEM and TEM. As shown in Figure 1a,b, the morphology of Bi_2_WO_6_ obtained without any Br precursor was observed to be in the form of hierarchical microspheres of ~2 μm diameter. The remaining samples that were grown with the use of Br^−^ ions were found to be nanoplatelets. The FE-SEM images of BW-C, BW-T, and BW-K are shown in Figure 1d,e,g,h, and Figure 1j,k, respectively. Furthermore, chemical element scanning analysis (Figure 1c,f,i,l) revealed the presence of bismuth, tungsten, and oxygen in the BW samples, whereas Br was observed in the BW-C, BW-T, and BW-K samples (Table 1). The TEM images were recorded to understand the detailed morphology of the prepared Bi_2_WO_6_ samples, as shown in Figure 2. Figure 2a,b shows that the BW sample formed regular microspheres during the hydrothermal reaction, whereas the Bi_2_WO_6_ samples that were synthesized with the assistance of Br^−^ ion precursors formed nanoplatelets (Figure 2c,d for BW-C, Figure 2e,f for BW-T, and Figure 2g,h for BW-K). According to a recent report [37], the initial presence of Bi_2_O_2_^2+^ layers is responsible for the growth of Bi_2_WO_6_ nanoplatelets during hydrothermal reactions. Here, the crystallization exhibits the following two stages. In the first stage, complexes of the Bi_2_O_2_^2+^ layer interact with WO_4_^2−^ tetrahedral units, leading to the stacking of WO_4_^2−^ in between the Bi_2_O_2_^2+^ layers. This interaction initiates the growth of the disordered Bi_2_WO_6_ crystalline phase. As the reaction advances, the disordered Bi_2_O_2_^2+^ stacks more WO_4_^2−^ tetrahedral units between the layers of Bi_2_O_2_^2+^ [38]. In the second step, this disordered phase evolves into Bi_2_WO_6_ nanoplatelets by “sideways” attachment along the *ac* plane.

### 3.2. Structural Characterization

To determine the crystalline phase and purity of the prepared Bi_2_WO_6_ nanostructures, powder X-ray diffraction patterns were recorded (Figure 3). As shown in Figure 3, the BW sample exhibited peaks at diffraction angles of 28.2°, 32.8°, 47.0°, 55.8°, 58.4°, 68.8°, 75.8°, and 78.3°, corresponding to the (131), (200), (202), (133), (262), (400), (391), and (402) planes, respectively, of the orthorhombic Bi_2_WO_6_ crystalline phase with a space group of Pbca (JCPDS No: 00-039-0256); no other crystalline phases were observed. The diffraction patterns of the remaining samples that were prepared by adding CTAB, TBAB, and KBr (BW-C, BW-T, and BW-K) during the synthesis also showed similar diffraction peaks, confirming Bi_2_WO_6_ formation [39]. The intensity of the diffraction planes was significantly altered due to the adsorption of Br^−^ ions during the crystal growth, which would control the stacking of layers in Bi_2_WO_6_ [36]. Among the samples, Bi_2_WO_6_ obtained using TBAB exhibited the highest crystallinity. 

Figure 4 illustrates the micro-Raman spectra of BW NSs that were prepared via a one-step hydrothermal process in the presence of various Br^−^ ion precursors. According to Figure 4, all the Raman bands of the BW sample can be assigned to the orthorhombic Bi_2_WO_6_ crystal phase. The prepared BW samples exhibited Raman vibrational bands at 122, 165, 267, 288, 312, 424, 720, 800, and 831 cm^−1^. The peaks positioned at 288 and 312 cm^−1^ in Figure 4 are characteristic of the Bi_2_WO_6_ orthorhombic phase and originate from the *E_g_* bonding modes [40]. 

The 312 cm^−1^ peak is assigned to the translation modes involving the simultaneous motions of Bi^3+^ and WO_6_^6−^ ions. The peaks observed at 720, 800, and 831 cm^−1^ correspond to the symmetric and asymmetric stretching of WO_6_ [41,42]. Another peak at 424 cm^−1^ is associated with the bending mode of WO_6_. In comparison to the BW sample, an extra peak around 122 cm^−1^ was observed in the BW-C, BW-T, and BW-K samples, representing the formation of Bi-Br bond vibrations, which are a consequence of Br-containing BW NSs [43,44]. Thus, it was concluded that extra Bi-Br bonds were formed. 

The chemical compositions and oxidation states of the constituent elements in the prepared Bi_2_WO_6_ nanostructures were studied via X-ray photoelectron spectroscopy (XPS). As shown in Figure 5, the XPS survey scan spectra of the BW sample displayed peaks related to Bi, W, and O. Furthermore, due to the presence of Br ions in the BW-C, BW-T, and BW-K samples, their survey scan spectra displayed Br-3d peaks, in addition to Bi, W, and O elements. As shown in Figure 6a, the binding energy values in the BW samples at 159.10 and 164.42 eV were attributed to the Bi-4f_7/2_ and Bi-4f_5/2_, respectively, of Bi^3+^, with a well-separated spin-orbit component of Δ = 5.3 eV [45]. Moreover, the high-resolution spectrum of W-4f shown in Figure 6b exhibited peaks at 35.30 and 37.47 eV (with Δ = 5.3 eV) corresponds to the W-4f_7/2_ and W-4f_5/2_ energy levels, respectively, with an oxidation state of +6 [20]. Figure 6c presents the O-1s spectrum of the BW sample. As shown, the deconvolution spectrum shows two peaks around 530.0 and 531.0 eV, which are characteristic of lattice oxygen and surface-adsorbed oxygen from the atmosphere, respectively [46,47]. The high-resolution spectra of other BW samples are also included for comparison and clearly show a shift in the binding energy values. Figure 6d provides the high-resolution Br-3d spectra originating from the Br^−^ ion-based precursors used to prepare Bi_2_WO_6_ ultrathin nanosheets. As shown, the BW sample did not exhibit any Br-3d spectrum, whereas the remaining samples showed well-resolved peaks around 67.40 and 68.44 eV for BW-C, 67.90 and 68.94 eV for BW-T, and 68.05 and 69.10 eV for BW-K [35]. These peaks indicate that the Br^−^ ions from CTAB, TBAB, and KBr were bonded to the surface Bi atoms in the layered structure. Consequently, the morphology was converted to that of typical ultrathin nanosheets. Notably, the XPS signals of Bi, W, and O in Figure 6 shifted toward lower binding energies after adding Br-based precursors to the reaction medium. This phenomenon was due to the formation of chemical bonds between Br and Bi_2_WO_6_ (Bi-O-Br), resulting in a change in the charge distribution in the host component, Bi_2_WO_6_. Because Bi_2_WO_6_ has a layered perovskite structure consisting of alternating [Bi_2_O_2_]^2+^ and [WO_4_]^2−^ layers, bromine is conducive to being inserted into the interlayer [38,48]. The interaction between the Br^−^ ions and Bi_2_WO_6_ caused the XPS peaks to shift toward a lower binding energy.

To understand the textural and porous structure of the prepared BW NSs, the N_2_ adsorption-desorption isotherms and their corresponding pore size distribution plots were measured (Figure 7). All curves shown in Figure 7 exhibit a type-IV isotherm with a narrow hysteresis loop, representing mesoporous structures. From the BET calculations, the specific surface area values of the obtained samples were determined as 18.8, 23.0, 32.4, and 35.4 m^2^/g for BW, BW-C, BW-T, and BW-K, respectively. From these measurements, it was confirmed that the morphological changes from microspheres (BW) to nanosheets (BW-C, BW-T, and BW-K) facilitated a higher surface area, which is highly beneficial for improving photocatalytic activity [49].

### 3.3. Optical Properties and Band Structure Studies

To determine the light-harvesting nature of the synthesized Bi_2_WO_6_ samples, UV-vis absorption spectra were recorded, as presented in Figure 8a. The plots in Figure 8a show that all the BW samples exhibited weak visible-light absorption, with an absorption band edge of approximately 450 nm and slight changes. These slight changes in the absorption edges of the Bi_2_WO_6_ samples are due to the change in morphology from microspheres to nanoplatelets. The bandgap energies of the prepared Bi_2_WO_6_ samples were determined using the absorbance data shown in Figure 8a, and their corresponding Tauc plots [50] are shown in Figure 8b. As shown, the bandgap energies of the BW, BW-C, BW-T, and BW-K samples are 2.80, 2.70, 2.79, and 2.66 eV, respectively. As discussed in Section 3.1, the adsorbed Br^−^ ions on the surface reduce the bandgap energy of the nanoplatelets. The Bi atoms were coordinately unsaturated on the surface, resulting in layered surfaces that exposed a large number of reactive sites (i.e., an increased specific surface area, as shown in Figure 7). Under visible-light irradiation, holes are directly produced on the surfaces and electrons in the middle layer, leading to efficient charge separation. Both the improved specific surface area and efficient charge separation result in higher photocatalytic activity [36]. To calculate the valence band positions (*E_VB_*) of the prepared Bi_2_WO_6_ samples, valence band XPS (VB-XPS) spectra were recorded (Figure 8c). The *E_VB_* values of BW, BW-C, BW-T, and BW-K were determined to be 2.10, 1.55, 1.95, and 1.90, respectively. Measurement errors were removed using the following equation: *E_VB_* = *φ* + *VB_xps_* − 4.44 eV, where *φ* with 4.2 eV and 4.44 eV corresponds to the work-function of the XPS analyzer and the vacuum level, respectively [1]. Then, the final VB positions in the band structure of the Bi_2_WO_6_ system were calculated as 1.86, 1.31, 1.71, and 1.66 eV for BW, BW-C, BW-T, and BW-K respectively. Furthermore, by using Equation (1) [51],
*E_CB_ = E_VB_ − E_g_*(1)
the corresponding conduction band positions (E*_CB_*) were calculated as −0.94, −1.39, −1.08, and −1.0 eV, respectively, versus NHE. The relative arrangements of *E_VB_* and *E_CB_* are shown in Figure 8d. As shown, the calculated VB positions of the prepared Bi_2_WO_6_ samples are above (less positive) the OH^−^/•OH potential (1.99 eV vs. NHE) and, hence, may not produce the •OH radicals; conversely, the CB values are more negative than O_2_/•O_2_^−^ potential (−0.33 eV vs. NHE), demonstrating that the electrons in CB of Bi_2_WO_6_ have the ability to form •O_2_^−^ radicals, which participate in the photocatalytic reactions to degrade environmental pollutants [52].

To determine the electron-hole recombination rate of the prepared photocatalytic systems, room-temperature steady-state photoluminescence (PL) spectra were obtained. Generally, a photocatalytic system that exhibits a lower PL intensity has a low electron-hole recombination rate; therefore, higher photocatalytic activity is observed [53]. The PL spectra were recorded at an excitation wavelength of 300 nm, as shown in Figure 9. All the PL spectra of the BW samples show similar features, and the emission band centered at approximately 465 nm is characteristic of the band-to-band transition seen in Bi_2_WO_6_. As shown in Figure 9, the PL intensity quenching in the BW-K sample indicates a decreased charge recombination rate, which improves the photocatalytic activity.

### 3.4. Photocatalytic Degradation of Tetracycline Pollutant Using BW NSs

The photocatalytic activities of the BW NS photocatalysts, derived from a hydrothermal reaction with and without the use of Br^−^ ion precursors in the reaction medium were tested toward the tetracycline (TC) degradation reaction using a 150 W Xe lamp (λ = 420–800 nm). Primary experiments revealed that the adsorption equilibrium of the samples was reached in ~30 min during the dark-adsorption stage. Moreover, both the light source and photocatalyst samples are essential for tetracycline degradation. In all the degradation experiments, a 20 ppm TC aqueous solution was used and irradiated for 100 min. 

According to Figure 10a, under visible-light irradiation, the BW sample exhibited moderate photocatalytic activity with a TC removal rate of only 71.0%, whereas the BW samples synthesized using Br^−^ ion precursors, i.e., BW-C, BW-T, and BW-K, showed increased photocatalytic TC degradation activities of 75.0, 78, and 84%, respectively, within 100 min, indicating that the morphology changes from microspheres to nanoplatelets improved the photocatalytic TC degradation activity. The improvement in the photocatalytic TC degradation activity of BW nanoplatelets compared to that of BW microspheres is attributed to the increased specific surface area and efficient charge separation properties. As discussed in Section 3.2, during the photocatalytic reactions, holes are produced on the surface, whereas electrons are produced in the middle layers of the BW nanoplatelets. As a result, improved charge separation and, consequently, higher photocatalytic activity were observed. The photocatalytic degradation kinetics of TC was studied using a pseudo-first-order model with the following formula [54]:ln(*C_o_/C*) = *kt*
(2)

The estimated rate constants for BW, BW-C, BW-T, and BW-K are 0.0117, 0.0129, 0.0149, and 0.0178 min^−1^, respectively (Figure 10b). Among the samples, BW-K showed the highest photocatalytic TC degradation rate constant, which was 1.52-, 1.38, and 1.19 times higher than those of the BW, BW-C, and BW-T samples, respectively, indicating the superiority of these samples [55]. From Appendix A and Figure 10a,b, it can be seen that the photocatalytic activity of the BW-K photocatalyst derived using KBr in the reaction medium is beneficial, compared to that of the reported photocatalysts, signifying its prospective application in degrading pharmaceutical antibiotic pollutants in wastewater. The durability of the BW-K sample was evaluated by conducting cycling tests. Moreover, the TC degradation activity of the BW-K sample increased with increasing catalyst loading (5, 10, 15, and 20 mg) (Figure 10c). After 100 min of visible-light irradiation, the degradation efficiencies were approximately 35.2, 58.6, 84.0, and 62.5%. This result can be explained as an increase in catalyst loading, which offers a large number of active sites for the TC degradation reaction. However, beyond a certain limit, an excessive amount of photocatalyst in the photochemical reaction can cause a shielding effect and may scatter light, resulting in lower photocatalytic efficiency. As shown in Figure 10d, the degradation rate remained at 83% even after five cycles, demonstrating that there was no apparent decrease in photocatalytic TC degradation and confirming the excellent durability of the photocatalyst sample. In order to elucidate this, FE-SEM and TEM images of the used photocatalyst sample were recorded and are given in Appendix A. As seen in Appendix A, there are no significant morphological changes observed, suggesting the stability of the BW-K sample. Based on these results, we provide a comparison table (Appendix A) to highlight the significance of our Bi_2_WO_6_-based nanostructures with the reported works. 

### 3.5. Photocatalytic Tetracycline Degradation Mechanism 

To explore the photocatalytic TC degradation mechanism of the BW-K photocatalyst, it is essential to determine the role of photoinduced electrons and holes during photocatalytic reactions [56]. Several chemical quenching experiments were conducted to identify the main reactive species involved in the photodegradation of TC. During the photocatalytic degradation of TC, benzoquinone (BQ), isopropanol (IPA), and ethylenediamine tetraacetic acid disodium salt (EDTA-2Na) were added separately in each experiment as trapping agents for superoxide radicals (•O_2_^−^), hydroxyl radicals (•OH), and holes (h^+^), respectively [57]. The corresponding results are provided in Figure 11. The TC degradation efficiency of the BW-K sample remained at 83.5% after the addition of IPA to the reaction medium. The experimental observations revealed that •OH radicals were not the main active species in the reaction. Compared to the BW-K sample without trapping/quenching agents, the photocatalytic TC degradation efficiency of the BW-K sample with added BQ was significantly quenched, indicating that •O_2_^−^ was the main reactive species in the photocatalytic degradation reaction. Furthermore, as shown in Figure 11, the degradation efficiency of the BW-K sample with the scavenger EDTA-2Na decreased slightly, implying that h^+^ played a minor role in the TC degradation reaction.

Based on the above results and band structure studies, a possible reaction pathway for TC degradation over Bi_2_WO_6_ (BW-K) is proposed in Figure 2. Under visible-light irradiation, the electrons in the VB are excited to the CB of Bi_2_WO_6_, while the holes remain in the VB. The CB position of Bi_2_WO_6_ is more negative than the O_2_/•O_2_^−^ potential (−0.33 eV vs. NHE); hence, the electrons have high power to convert the adsorbed O_2_ to •O_2_^−^ radicals, and subsequently, degrade the TC molecules [58]. Meanwhile, the holes present in the VB (+1.66 eV) of Bi_2_WO_6_ are not positive enough to oxidize OH^−^ into •OH radicals. Therefore, the remaining holes in the VB of Bi_2_WO_6_ react with the adsorbed TC molecules and form byproducts. Finally, in the Bi_2_WO_6_ photocatalytic system, the active species of •O_2_^−^ radicals (major active species) and h^+^ (minor active species) could oxidize the adsorbed tetracycline molecules into CO_2_ and H_2_O. 

After bonding the Br^−^ in the layered structure of Bi_2_WO_6_, the morphology was clearly changed from bulk microspheres to nanoplates, the surface area was improved to present more surface reactive sites, and the band gap was decreased for greater visible light absorption performance. The excellent photocatalytic TC degradation activity of the BW-K sample is probably related to the increased surface-based active sites and visible-light absorption by the insertion of Br^−^ into the interlayer of orthorhombic Bi_2_WO_6_. The unique structure of BW-K could facilitate the interfacial charge separation and transfer it to the catalyst surface for degradation reactions more proficiently, as confirmed by PL measurements. Furthermore, the insertion of bromine into the layer structure of Bi_2_WO_6_ shifted the VB and CB potentials to increase the reduction power of the photogenerated electrons, which, together with a favorable structure for separation and the transfer of carriers, contributed to the boosted degradation of tetracycline in water.

## 4. Conclusions

Orthorhombic Bi_2_WO_6_ nanostructures with different morphologies were effectively prepared using a facile, single-step hydrothermal process. The present bottom-up bromine ion-assisted hydrothermal synthesis demonstrates that the presence of a few Br^−^ ions in Bi_2_WO_6_ is responsible not only for the growth of nanoplates but also for its morphology tuning from bulk microspheres to nanoplates. During the growth of Bi_2_WO_6_, the adsorption of Br^−^ ions into the network prevents the assembly of several layers, resulting in a nanoplate morphology and a narrower bandgap compared to bulk Bi_2_WO_6_. The obtained Bi_2_WO_6_ nanoplates exhibited a sandwich structure of [BiO]^+^–[WO4]^2−^–[BiO]^+^ with Br-Bi-Bi-Br stacking. The resulting nanoplate structure effectively suppressed the charge recombination during the photocatalytic reactions, resulting in enhanced photocatalytic TC degradation. Moreover, the nanoplate structure of Bi_2_WO_6_ offers improved light absorption and a higher number of reactive sites for oxidation. The unique growth of Bi_2_WO_6_ nanoplates via a route mediated by the Br-ions provides an effective strategy for realizing efficient photocatalysts for advanced oxidation processes, and the same rational design approach can be extended to other layered photocatalytic systems. 

## Data Availability

The data presented in this study are available on request from the corresponding author.

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
