# Peer review of "Bromine Ion-Intercalated Layered Bi2WO6 as an Efficient Catalyst for Advanced Oxidation Processes in Tetracycline Pollutant Degradation Reaction"

_nanomaterials, 2023, doi:10.3390/nano13182614_

Round 1
Reviewer 1 Report
In this manuscript, the authors reported the use of bromine-ion to optimize the morphology of Bi2WO6 and increase the formation of active sites on the photocatalyst surface. This manuscript is clearly written and reasonably analyzed. The reviewer thinks this manuscript is interesting and can be published in Nanomaterials after clarifying the following issues.
(1) The authors suggest Bromine ions are intercalated in layered Bi2WO6, but they also state the adsorption of Bromine ions on the Bi2WO6 surface in the manuscript. In what form does Bromine exists in Bi2WO6? Please clarify it.
(2) The authors are suggested to discuss why Br results in the formation of Bi2WO6 nanosheets.
(3) In Fig. 8b, the scale of the y-axis should be shown (starting from zero) since it is important for the correct determination of the intercept of the x-axis.
(4) In the y-axis of Fig. 10c and d, is there any difference between degradation rate and degradation efficiency?
(5) In Scheme 2, the authors are suggested to illustrate the photocarrier separation mechanism by Br.
(6) The two related papers are suggested to read and mentioned (Mater. Res. Bull. 149 (2022) 111711, Appl. Surf. Sci. 611 (2023) 155681).

The English language is acceptable.
Author Response
Replies to Reviewer comments
Manuscript ID: nanomaterials-2617715
Type: Article
Title: Bromine ions-intercalated layered Bi2WO6 as an efficient catalyst for advanced oxidation processes in tetracycline pollutant degradation reaction
Reviewer’s comments
Reviewer #1
(1) The authors suggest Bromine ions are intercalated in layered Bi2WO6, but they also state the adsorption of Bromine ions on the Bi2WO6 surface in the manuscript. In what form does Bromine exist in Bi2WO6? Please clarify it.
Reply: Thank you for our comment regarding the state of Br ions in the present work. In this work, we attempted to change the morphology of Bi2WO6 from bulk microspheres to nanoplates by using Br-containing precursors which were used during the synthesis of Bi2WO6. We are aware of the fact that the size and morphology play a major role in enriching the photocatalytic activity. Changes in the morphology of Bi2WO6 can increase the exposure of reactive sites, thus increasing pollutant degradation activity. Therefore Br-containing precursors were used during the synthesis of Bi2WO6. As per your comment, at initial stage of crystal growth, the Br− ions from CTAB/TBAB/KBr strongly adsorb on the layered surface and block the further stacking of layers, resulting in nanoplates instead of bulk microspheres.
Next, we have explained the state/form of Br from the high-resolution XPS spectra of Br-3d spectra (in page 10, Figure 6d). As shown, the sample BW did not exhibit any Br-3d spectrum, whereas the remaining samples showed well-resolved peaks around 67.40 and 68.44 eV for BW-C, 67.90 and 68.94 eV for BW-T; and 68.05 and 69.10 eV for BW-K. These peaks indicate that the Br− ions from CTAB, TBAB, and KBr were bonded to the surface Bi atoms in the layered structure. Consequently, the morphology was converted to that of typical ultrathin nanosheets. Notably, the XPS signals of Bi, W, and O in Figure 6 shift toward lower binding energies after adding Br-based precursors to the reaction medium. This phenomenon was due to the formation of chemical bonds between Br and Bi2WO6 (Bi-O-Br), resulting in a change in the charge distribution in the host component Bi2WO6. Because Bi2WO6 has a layered perovskite structure consisting of alternating [Bi2O2]2+ and [WO4]2- layers, bromine is conducive to being inserted into the interlayer. The interaction between the Br− ions and Bi2WO6 caused the XPS peaks to shift toward a lower binding energy.
(2) The authors are suggested to discuss why Br results in the formation of Bi2WO6 nanosheets.
Reply: As per your suggestion, the role of Br in the formation of Bi2WO6 nanosheets is discussed as follows (page 5, Line 184-196). The BW sample was formed as regular microspheres during the hydrothermal reaction, whereas the Bi2WO6 samples synthesized with the assistance of Br− ion precursors were formed as nanoplatelets (Figure 2c, d for BW-C; Figure 2e, f for BW-T and Figure 2g, h for BW-K). According to a recent report, the initial presence of Bi2O22+ layers is responsible for the growth of Bi2WO6 nanoplates during hydrothermal reactions. Here, the crystallization is ascribed to the following two stages: In the first stage, complexes of Bi2O22+ layer interact with WO42− tetrahedral leading to the stacking of WO42− in between the Bi2O22+ layers. This interaction initiated the growth of the disordered Bi2WO6 crystalline phase. As the reaction advances, the disordered Bi2O22+ stacks more WO42− tetrahedral units between the layers of Bi2O22+. In the second step, these disordered phase evolves into Bi2WO6 nanoplatelets by “sideways” attachment along the ac plane.
(3) In Fig. 8b, the scale of the y-axis should be shown (starting from zero) since it is important for the correct determination of the intercept of the x-axis.
Reply: As per your suggestion, Y-axis scaling starting from zero is provided in the revised manuscript.
(4) In the y-axis of Fig. 10c and d, is there any difference between degradation rate and degradation efficiency?
Reply: We are sorry for the typo error. In the revised manuscript, both Figures 10c and d are provided with degradation efficiency (%). Both units are the same. Thanks for highlighting our mistake.
(5) In Scheme 2, the authors are suggested to illustrate the photocarrier separation mechanism by Br.
Reply: In the present work, we attempted to change the morphology of Bi2WO6 from bulk microspheres to nanoplates by using Br-containing precursors during the reaction. We are aware of the fact that the size and morphology play a major role in enriching the photocatalytic activity. Changes in the morphology of Bi2WO6 can increase the exposure of reactive sites, thus increasing pollutant degradation activity. Therefore, at the initial stage of crystal growth, the Br− ions from CTAB/TBAB/KBr strongly adsorb on the layered surface and block the further stacking of layers, resulting in nanoplates instead of bulk microspheres. Moreover, the structural characterization reveals that Br species are bonded to Bi and O atoms a Bi-O-Br. Here the nanoplates-like structure of Bi2WO6 is open and can strongly interact with the Br- ions. Moreover, the effective charge separation in BW-K is attributed to the unique morphology of the Bi2WO6 layered structure. The Br-bonding with Bi and O atoms in the layered structure can effectively increase the charge separation and it is very difficult to illustrate in the mechanism. We hope the Reviewer agrees with our statement. The successfully synthesized layered Aurivillius oxide Bi2WO6 has a sandwich substructure of [BiO]+–[WO4]2−–[BiO]+, mimicking heterojunction interface with space charge allows the efficient charge separation during photochemical reactions. A similar charge separation mechanism (excluding Br) provided in the current manuscript is also supported by previous works.
(6) The two related papers are suggested to read and mentioned (Mater. Res. Bull. 149 (2022) 111711, Appl. Surf. Sci. 611 (2023) 155681).
Reply: We have carefully read both published works and cited as References [45] and [47].

Reviewer 2 Report
This manuscript reported the synthesis of bromine ions-intercalated layered Bi2WO6, which exhibited enhanced photocatalytic activity towards tetracycline pollutant degradation. However, the quality of the manuscript needs to be improved. The following issues can be considered in revising the manuscript.
(1) There are 4 nitrogen adsorption-desorption isotherms. The four figures (a, b, c and d) can be merged into ONE figure for easy comparison.
(2) Fig. 10 c is the stability, and Fig. 10 d is the effect of catalyst dosage on TC removal rate. In my opinion, the optimized dosage should be determined, and then to study the photocatalytic stability. So, Fig. 10d should be presented first, and then Fig. 10c. So, sequence of Fig. 10c and Fig. 10d should be replaced.
(3) Although radical quenching experiments were carried out (Fig. 11), it is suggested that the radical trapping by ESR technique should be presented.
(4) In preparation of Br-ion intercalated Bi2WO4, surfactants such as CTAB/cetyltrimethylammonium bromide (CTAB)/tetrabutylammonium bromide (TBAB) were added. To illustrate the effect of surfactants on the structure and photocatalytic activty of Bi2WO6, control experiments should be perfromed. (1) Bi2WO6 prepared in CTAB/cetyltrimethylammonium bromide (CTAB)/tetrabutylammonium bromide (TBAB) without KBr; (2) Bi2WO6 prepared in KBr but without CTAB/cetyltrimethylammonium bromide (CTAB)/tetrabutylammonium bromide (TBAB).
(5) Some papers related to Bi2WO6 can be referenced such as
In-situ transformation of Bi2WO6 to highly photoreactive Bi2WO6@Bi2S3 nanoplate via ion exchange, Chin. J. Catal. 2018, 39, 718-727.
Author Response
Replies to Reviewer comments
Manuscript ID: nanomaterials-2617715
Type: Article
Title: Bromine ions-intercalated layered Bi2WO6 as an efficient catalyst for advanced oxidation processes in tetracycline pollutant degradation reaction
Reviewer’s comments
Reviewer #2
(1) There are 4 nitrogen adsorption-desorption isotherms. The four figures (a, b, c and d) can be merged into ONE figure for easy comparison.
Reply: Thank you for your suggestion. As per your direction, we have modified as single figure. The following Figure was inserted as Fig. 7 in the revised manuscript.
Figure 7. N2 adsorption-desorption isotherms of Bi2WO6 photocatalyst samples.
(2) Fig. 10 c is the stability, and Fig. 10 d is the effect of catalyst dosage on TC removal rate. In my opinion, the optimized dosage should be determined, and then to study the photocatalytic stability. So, Fig. 10d should be presented first, and then Fig. 10c. So, sequence of Fig. 10c and Fig. 10d should be replaced.
Reply: Thank you for commenting on the sequence of figures in Figure 10. After checking the data, we also felt the same and modified. Figure 10 c and Figure 10d replaced each other.
Figure 10. (a) Photocatalytic degradation of TC over the Bi2WO6 samples under visible-light irradiation, (b) Kinetic curves and rate constants for photodegradation of TC over the as-prepared samples, (c) Catalyst dosage experiments and (d) Recycling tests of optimized BW-K sample. Test conditions are maintained as λ=420-800 nm; TC concentration 20 ppm; catalyst dosage as 15 mg.
(3) Although radical quenching experiments were carried out (Fig. 11), it is suggested that the radical trapping by ESR technique should be presented.
Reply: Thank you very much for your comment regarding the radical trapping experiments by ESR technique. We agree with the Reviewer that the ESR radical trapping experiments will be useful and supplement the radical quenching experiments. Our university doesn’t have ESR instrument facilities and we also enquired other institutes and rare to find. Though, we find one institution and it needs nearly two months for measurements. At present we are unable to provide the data, however, we will consider your suggestion for our next works. We hope the Reviewer will agree to the current situation.
(4) In preparation of Br-ion intercalated Bi2WO4, surfactants such as CTAB/cetyltrimethylammonium bromide (CTAB)/tetrabutylammonium bromide (TBAB) were added. To illustrate the effect of surfactants on the structure and photocatalytic activty of Bi2WO6, control experiments should be perfromed. (1) Bi2WO6 prepared in CTAB/cetyltrimethylammonium bromide (CTAB)/tetrabutylammonium bromide (TBAB) without KBr; (2) Bi2WO6 prepared in KBr but without CTAB/cetyltrimethylammonium bromide (CTAB)/tetrabutylammonium bromide (TBAB).
Reply: Thank you for your comment on the synthesis of Bi2WO6 nanostructures. Earlier, we have already performed some controlled experiments by using two types of Br precursors without the other. The obtained samples are not good enough for photocatalytic applications. We feel that the reaction conditions may not have an impact in the formation of effective Bi2WO6 nanostructures. Therefore we have not included it in the manuscript.
(5) Some papers related to Bi2WO6 can be referenced such as “In-situ transformation of Bi2WO6 to highly photoreactive Bi2WO6@Bi2S3 nanoplate via ion exchange, Chin. J. Catal. 2018, 39, 718-727.”
Reply: As per the Reviewer and academic editor's suggestion, the results of above above-published work is cited in the revised manuscript (Reference [14]).

Reviewer 3 Report
Nanomaterials
Manuscript ID: nanomaterials-2617715
Title: Bromine ions-intercalated layered Bi2WO6 as an efficient catalyst for advanced oxidation processes in tetracycline pollutant degradation reaction
I think that is a very interesting study well organized and presented, however, some issues should be improved. Thus, I recommend a publication of this study to Nanomaterials journal after the authors consider the following major revisions.
Comment #1
The authors need to improve on their bibliography. I think that they should add some references in order to enrich the introduction section in terms of the active sites of the presented catalyst. The authors could add more information in terms of the presented active sites O2– species and holes (h+) that can be produced on the catalyst surface participating in the oxidation process (e.g., oxidative coupling of methane). Please include some of the most important findings of the following research studies in the beginning of the introduction section (lines 31-33).
[1]. G.I. Siakavelas, N.D. Charisiou, A. AlKhoori, S. Gaber, V. Sebastian, S.J. Hinder, M.A. Baker, I.V. Yentekakis, K. Polychronopoulou, M.A. Goula, Oxidative coupling of methane on Li/CeO2 based catalysts: Investigation of the effect of Mg- and La-doping of the CeO2 support. Molecular Catalysis 520 (2022) 112157.
[2]. G.I. Siakavelas, N.D. Charisiou, A. AlKhoori, V. Sebastian, S.J. Hinder, M.A. Baker, I.V. Yentekakis, K. Polychronopoulou, M.A. Goula, Cerium oxide catalysts for oxidative coupling of methane reaction: Effect of lithium, samarium and lanthanum dopants. Journal of Environmental Chemical Engineering 10 (2022) 107259.
Furthermore, please improve the innovation of your study in the introduction section, and what are the new aspects being introduced on this research topic?
Comment #2
Please provide some figure (SEM or TEM) with the catalytic sample after catalytic activity experiment in order to provide more information about the stability of the structure. Is there any structural change?
Author Response
Replies to Reviewer comments
Manuscript ID: nanomaterials-2617715
Type: Article
Title: Bromine ions-intercalated layered Bi2WO6 as an efficient catalyst for advanced oxidation processes in tetracycline pollutant degradation reaction
Reviewer’s comments
Reviewer #3
Comment #1
The authors need to improve on their bibliography. I think that they should add some references in order to enrich the introduction section in terms of the active sites of the presented catalyst. The authors could add more information in terms of the presented active sites O2– species and holes (h+) that can be produced on the catalyst surface participating in the oxidation process (e.g., oxidative coupling of methane). Please include some of the most important findings of the following research studies in the beginning of the introduction section (lines 31-33).
[1]. G.I. Siakavelas, N.D. Charisiou, A. AlKhoori, S. Gaber, V. Sebastian, S.J. Hinder, M.A. Baker, I.V. Yentekakis, K. Polychronopoulou, M.A. Goula, Oxidative coupling of methane on Li/CeO2 based catalysts: Investigation of the effect of Mg- and La-doping of the CeO2 support. Molecular Catalysis 520 (2022) 112157.
[2]. G.I. Siakavelas, N.D. Charisiou, A. AlKhoori, V. Sebastian, S.J. Hinder, M.A. Baker, I.V. Yentekakis, K. Polychronopoulou, M.A. Goula, Cerium oxide catalysts for oxidative coupling of methane reaction: Effect of lithium, samarium and lanthanum dopants. Journal of Environmental Chemical Engineering 10 (2022) 107259.
Furthermore, please improve the innovation of your study in the introduction section, and what are the new aspects being introduced on this research topic?
Reply: Thank you for your comment regarding the reactive species on the catalyst surface. As per your suggestion, we have thoroughly read the above-published results and incorporated the important results in the revised manuscript. The importance and significance of the current work is highlighted in the introduction part (lines from 85-113).
Ultrathin nanosheets with a thickness of a few nanometers have been considered excellent photocatalytic materials because of their rapid mass transport, high charge transfer rate, and the abundance of active sites for catalytic reactions [32]. Unfortunately, converting the morphology of the bulk structure into ultrathin nanosheets is challenging because the intralayer coordination bonds are easily broken during synthesis. Wet bottom-up chemical synthesis has been demonstrated to be the most favorable approach for preparing ultrathin nanosheets with high yields [33,34]. It should also be noted that this synthesis protocol prevents the grown ultrathin nanosheets from stacking together. Therefore, in this study, we propose a single-step hydrothermal method for the preparation of Bi2WO6 nanostructures and detail the importance of surfactants that alter the morphology from bulk hierarchical microspheres to ultrathin nanosheets and further photophysical changes. These morphological changes introduced via the organic surfactant-mediated synthesis route were found to significantly improve the photocatalytic activity by lowering the bandgap, increasing the specific surface area, and improving the charge separation properties.
In this study, we used a bromine-containing precursor (cetyltrimethylammonium bromide (CTAB)/ tetrabutylammonium bromide (TBAB)/potassium bromide (KBr)) assisted bottom-up synthesis to obtain ultrathin nanosheets. The successfully synthesized ultrathin Aurivillius oxide Bi2WO6 has a layered structure. Moreover, Br− ions from CTAB/TBAB/KBr strongly adsorb on the layered surface and block the stacking of layers. Ultrathin Bi2WO6 nanosheets were obtained whereas the Bi2WO6 microspheres were observed without the addition of Br-containing precursors. As a result, the Br− ions on the surface reduce the bandgap in ultrathin nanosheets. These Bi2WO6 nanosheets possess a lower bandgap and larger specific surface area than the Bi2WO6 microspheres. Further, we speculated that the introduced Br− ions might reduce the energy barrier in oxygen vacancy (OV) generation, thus resulting in the gradient concentration of OVs on different exposed facets of materials [30, 35]. The photocatalytic activity of the obtained Bi2WO6 nanostructures was determined by the photocatalytic degradation of the pharmaceutical pollutant tetracycline hydrochloride under visible-light irradiation.
Comment #2
Please provide some figure (SEM or TEM) with the catalytic sample after catalytic activity experiment in order to provide more information about the stability of the structure. Is there any structural change?
Reply: As per your comment, we have recorded the SEM and TEM images of the optimized photocatalyst sample and provided in the experimental supporting file. The following images are provided in the revised file and there is no significant morphological changes were observed.
Figure S2. (a) FE-SEM and (b) TEM images of the BW-K photocatalyst sample after cycling reactions.
The relevant discussion was added at the line numbers 387-392 of the revised manuscript and Figure S2 in provided in the supporting information file.

Round 2
Reviewer 2 Report
The authors have carefully revised the mansucript according to the comments. However, the quality of Scheme 2 is poor. The dark background make it hard to get related information. Plese remove the dark background from the image.
Author Response
Comment: The authors have carefully revised the manuscript according to the comments. However, the quality of Scheme 2 is poor. The dark background makes it hard to get related information. Please remove the dark background from the image.
Reply: Thank you for accepting our revisions. As you suggested, we have removed the background of Scheme 2 in the revised manuscript.
Many thanks for recommending our manuscript for publication.
Reviewer 3 Report
Accept in present form
Author Response
Comment: Accept in present form
Reply: Many thanks to the Reviewer for accepting our manuscript for publication.